# Analysis of Clinical and Biochemical Parameters and the Effectiveness of Surgical Treatment in Patients with Primary Hyperparathyroidism: A Single-Center Study

**DOI:** 10.3390/jcm14030996

**Published:** 2025-02-04

**Authors:** Jakub Migoń, Michał Miciak, Dominika Pupka, Szymon Biernat, Łukasz Nowak, Krzysztof Kaliszewski

**Affiliations:** 1Department of General and Oncological Surgery, University of Zielona Góra, 65-046 Zielona Góra, Poland; j.migon@szpital.zgora.pl; 2Department of General Surgery, University Centre of General and Oncological Surgery, Wroclaw Medical University, 50-556 Wroclaw, Poland; szymon.biernat@student.umw.edu.pl; 3Department of General Surgery, Saint Hedwig’s of Silesia Hospital, 55-100 Trzebnica, Poland; dominika.pupka@gmail.com; 4Department of Minimally Invasive and Robotic Urology, University Centre of Excellence in Urology, Wroclaw Medical University, 50-556 Wroclaw, Poland; lukasz.nowak@umw.edu.pl

**Keywords:** parathyroid gland, primary hyperparathyroidism, endocrine surgery, hypercalcemia, single-center experience, open minimally invasive parathyroidectomy

## Abstract

**Background**: Primary hyperparathyroidism (PHPT) causes an imbalance of calcium-phosphate metabolism in the form of hypercalcemia and hypophosphatemia, leading to dysfunction in various organs. The main cause is a benign tumor of the parathyroid gland (adenoma), leading to excessive and uncontrolled secretion of parathyroid hormone (PTH). Difficulties in diagnosing PTHP are also compounded by the possibility of asymptomatic course at the early disease stages. The gold standard treatment involves removing the pathological gland, while pharmacological options are reserved for candidates ineligible for surgery. **Methods**: In our study, we assessed the effectiveness of surgical treatment and the factors influencing outcomes and complications by analyzing the records of 125 patients with PHPT who underwent parathyroidectomy at the University Centre of General and Oncological Surgery of the Wroclaw Medical University from 2008 to 2017. We considered sociodemographics, laboratory results, comorbidities, complications, procedure details, and outcomes. The procedures included 93 open minimally invasive parathyroidectomies (OMIPs), 11 unilateral neck explorations, and 21 bilateral neck explorations. **Results**: Single-gland pathology was common (101 patients), while 24 had involvement of two glands. The left inferior parathyroid gland was the most frequently affected (*n* = 65; 43.6%). Histopathological examination consistently indicated the presence of parathyroid adenoma in all patients. Complications following parathyroidectomy were observed in 11 (8.8%) patients. Calcium normalization after six months was observed in 119 patients (95.2%). The surgical technique, the location of the adenoma, and the sex and age of the patient did not appear to affect the effectiveness of treatment. **Conclusions**: Parathyroidectomy is highly effective in treating PHPT, irrespective of patient age, sex, or gland location. It leads to decreased serum PTH and total calcium levels while increasing the inorganic phosphate concentration.

## 1. Introduction

Hyperparathyroidism (HPT) is a pathological condition characterized by hyperactivity of the parathyroid glands, resulting in aberrantly elevated levels of parathyroid hormone (PTH) in the circulatory system. HPT can be categorized into primary, secondary, and tertiary hyperparathyroidism [1]. Primary hyperparathyroidism (PHPT) results from a dysregulated increase in PTH secretion by the parathyroid glands, leading to concurrent hypercalcemia, as the parathyroid cells exhibit diminished sensitivity to or insensitivity to the suppressive influence of hypercalcemia. PHPT is typically acquired and is associated with autonomous glandular hyperplasia, with a prevalence of 85% for a single adenoma, 15% for multiple adenomas or parathyroid hyperplasia, and 1% for parathyroid cancer [2]. The musculoskeletal and urinary systems are predominantly affected by PHPT, resulting in conditions such as osteoporosis, osteopenia, osteoarticular pain, pathological fractures, kidney and ureteral stones, pyonephrosis, and renal failure [2]. Neurological manifestations may also transpire, with a wide spectrum of symptoms that range from mild (depressive mood, fatigue, and sleep disturbances) to severe (anxiety disorders, depression, cognitive impairment, hallucinations, and delusions) and, in extreme cases, coma [3]. Additionally, PHPT may manifest with hypertension, cardiac arrhythmias, or treatment-resistant anemia. Individuals with PHPT are at an increased risk of peptic ulcer disease, acute pancreatitis, and biliary tract stones [4,5]. According to the Fifth International Workshop on Evaluation and Management of Primary Hyperparathyroidism, the diagnosis of PHPT relies on laboratory test results, which primarily reveal elevated levels of PTH and calcium in the blood. In cases where high PTH levels coexist with normocalcemia, diagnostic evaluation should be expanded to include measurements of calcidiol (25(OH)D3) and creatinine levels to rule out secondary hyperparathyroidism [6]. Imaging modalities play a pivotal role in guiding surgical interventions and serve as adjunctive tools in the diagnostic process of PHPT, assisting surgeons in the preoperative planning and identification of potentially affected parathyroid glands. Standard imaging modalities for parathyroid glands include computed tomography (CT), magnetic resonance imaging (MRI), ultrasonography (USG), and scintigraphy. On ultrasound, enlarged parathyroid glands appear as hypoechoic structures compared with the surrounding tissue. Pathologically altered parathyroid glands located around the thyroid or within its parenchyma are the easiest to identify. However, ultrasound is not very effective for identifying ectopically located parathyroid glands [7]. CT and MRI are characterized by high, similar effectiveness and are mainly used in the diagnosis of recurrent or persistent hyperparathyroidism [8]. In preoperative diagnostics for locating adenomas, scintigraphy with [99mTc]Tc-MIBI radioisotope is widely used. According to various authors, the sensitivity of this test ranges from 55% to 100%, but it decreases in cases of generalized hyperplasia. Parathyroid scintigraphy can be performed via two methods: the double-tracer technique or the dual-phase technique [9]. In complex cases where the outcomes of conventional imaging tests are inconclusive or previous surgical interventions have been unsuccessful, positron emission tomography/computed tomography (PET/CT) using [11C]C-methionine or [18F]F-choline can be considered. The mechanism behind the increased accumulation of radiotracers in parathyroid adenoma cells remains unclear, but it is suspected to be due to increased cell proliferation and elevated production of preProPTH (a precursor of PTH). This approach can be helpful in determining the location of small or multiple parathyroid adenomas [10,11,12]. In the management of severe hypercalcemia, the foundational tenet is the attainment of adequate hydration and loop diuretics deployment to stimulate diuresis and augment calcium excretion [13]. A non-surgical approach for PHPT treatment is limited for patients with a high risk of postoperative complications. Bisphosphonates, agents inhibiting osteoclast-dependent bone osteolysis, may be deemed necessary in select cases. While the efficacy of cinacalcet in the context of PHPT remains less established, its application is extended to patients exhibiting secondary or tertiary hyperparathyroidism. Cinacalcet potentiates the sensitivity of the calcium-sensing receptor (CaSR), resulting in diminished PTH secretion and commensurate reductions in calcium levels [14]. Recent studies have suggested that cinacalcet may also possess the potential to reduce the size of parathyroid adenomas; however, these findings require further investigation to confirm their validity [15]. Parathyroidectomy represents the contemporary gold standard in the therapeutic landscape of PHPT [16]. Nevertheless, this surgical intervention is not devoid of potential postoperative complications, including hemorrhage, postoperative wound infection, and recurrent laryngeal nerve palsy. The principal objective of this study was to conduct a retrospective analysis of the efficacy of surgical intervention in PHPT, elucidating the factors influencing the extent of the procedure. Additionally, the investigation sought to scrutinize the incidence of postoperative complications.

## 2. Materials and Methods

This single-center retrospective analysis involved 125 patients undergoing treatment for PHPT at the University Centre of General and Oncological Surgery of the Wroclaw Medical University from 2008 to 2017. Data were extracted from both paper and electronic medical records. The patients provided written informed consent for the analysis of their clinical data. Ethical approval was exempted by the local Ethics Committee of Wroclaw Medical University. A total of 32 patients were excluded from the study because of insufficient data. Patient qualification for surgical intervention involves the scrutiny of laboratory test results and a minimum of two imaging studies (ultrasonography, computed tomography, magnetic resonance, or parathyroid scintigraphy). Patient qualification for surgery was performed in the endocrinology and general surgery outpatient clinics. During the patient’s first visit, they were given questionnaires regarding mood (Personal Health Questionnaire Depression Scale, PHQ-8) and functional muscle strength (SARC-F). To assess osteoporotic changes, BMD (Bone Mineral Density) results and patient history of pathological fractures were evaluated. Assessment of cholelithiasis and urinary tract stones was based on the medical history regarding procedures performed for those conditions or on available imaging studies. Complaints such as nausea, headaches, and osteoarticular pain were analyzed based on the patient history records. Other nonspecific symptoms associated with disease progression included memory disorders, confusion, and skin itching. While the specific guidelines utilized were not documented, the indications for surgery were a clinical presentation of PHPT and a total calcium level above 10.50 mg/dL. Ninety-three open minimally invasive parathyroidectomies (OMIPs), 11 unilateral neck exploration procedures, and 21 bilateral neck exploration procedures were conducted, resulting in the removal of a total of 149 parathyroid glands. The classification of parathyroid gland locations was justified by their position relative to the recurrent laryngeal nerve. A successful parathyroidectomy was defined as an operation in which serum calcium levels did not exceed 10.50 g/dL during the six-month follow-up period. Patients were stratified into two age groups, adults (18–64 years) and older individuals (65 years and above), in accordance with the recommendations of the Central Statistical Office in Poland. The median patient age was 60 years, 56 for the female group and 61 for the male group. The classification of a surgeon as experienced was attributed to those who had performed more than 10 parathyroidectomies, whereas a less experienced surgeon was characterized by having conducted 10 or fewer parathyroidectomies. Hypophosphatemia severity was categorized as mild (3–2.60 mg/dL), moderate (2.50–1 mg/dL), or severe (<1 mg/dL) [14]. The normal values for total serum calcium and PTH in our study were 8.50–10.50 mg/dL and 10–60 pg/mL, respectively.

### Statistical Analysis

Statistical analysis was performed using PQStat software (PQStat Software, 2022, version 1.8.4.140) on quantitative and qualitative variables. Quantitative variables included basic descriptive statistics: mean value (M), standard deviation (SD), highest value, and lowest value. Qualitative variables included frequency and percentage. The normality of variable distributions was analyzed using the Kolmogorov–Smirnov test. The influence of variables and interaction effects was verified using the Sidak test. Relationships between qualitative variables were analyzed by cross-tabulation using the chi-square test, the chi-square reliability quotient, and Fisher’s exact test. For independent samples in non-parametric groups, the Mann–Whitney U test was applied. A significance level of *p* < 0.05 was considered statistically significant.

## 3. Results

The patient and clinical characteristics of the study group are provided in Table 1. In the study group, 43 (34.0%) patients had at least one comorbidity. The most common comorbidities were hypertension (21.6%) and diabetes (12.0%).

### 3.1. Analysis of PHPT Symptoms Occurrence According to Age, Calcium, and PTH Serum Levels

Patients aged 65 years and older significantly more often reported muscle weakness (98.1% vs. 51.3%; *p* < 0.05), urinary tract stones (86.7% vs. 33.3%; *p* < 0.05), osteoporosis (86.7% vs. 33.3%; *p* < 0.05), osteoarticular pain (86.7% vs. 31.9%; *p* < 0.05), and cholelithiasis (86.7% vs. 30.5%; *p* < 0.05). The relations between the occurrence of PHPT symptoms and age are presented in Table 2.

The relation between the occurrence of PHPT symptoms and total serum calcium levels is presented in Table 3. A significantly greater mean total serum calcium concentration was detected in patients reporting symptoms of muscle weakness (11.73 mg/dL vs. 11.03 mg/dL; *p* < 0.05), urinary tract stones (12.03 mg/dL vs. 11.02 mg/dL; *p* < 0.05), osteoporosis (12.05 mg/dL vs. 11.02 mg/dL; *p* < 0.05), osteoarticular pain (12.05 mg/dL vs. 11.03 mg/dL; *p* < 0.05), and cholelithiasis (12.03 mg/dL vs. 11.06 mg/dL; *p* < 0.05).

The symptoms of PHPT and their relation with PTH levels in the blood are presented in Table 4. Higher mean PTH levels in the blood were detected in patients reporting urinary tract stones (356.90 pg/mL vs. 296.42 pg/mL; *p* < 0.05), osteoporosis (356.03 pg/mL vs. 297.64 pg/mL; *p* < 0.05), and cholelithiasis (357 pg/mL vs. 298.52 pg/mL; *p* < 0.05).

### 3.2. Analysis of Perioperative Data and Surgical Findings in PHPT

The sex and the number of pathologically altered parathyroid glands did not influence the occurrence of clinical symptoms. The hospitalization duration for patients younger than 65 years ranged from 3 to 10 days (M = 3.90; SD = 1.5), whereas for those aged 65 and older, it ranged from 3 to 16 days (M = 5.16; SD = 2.3), with a statistically significant difference (*p* < 0.05). Patients who experienced postoperative complications had a significantly longer hospitalization period (M = 9.3; SD = 2.9) than did those without complications (M = 3.92; SD = 0.96) (*p* < 0.05). The surgeon’s level of experience did not affect the length of the hospital stay, which averaged 4.40 days (*p* = 0.920). A total of 149 parathyroid glands were excised during parathyroidectomy procedures. Complications following parathyroidectomy were observed in 11 (8.8%) patients within the study cohort. Among these, the most common complications included postoperative wound infection (*n* = 4; 36%) and postoperative bleeding (*n* = 3; 27%). Paresthesia (*n* = 2; 18%) and tetany (*n* = 1; 9%) occurred less frequently. Cardiopulmonary failure was observed in one (9%) patient during the postoperative period. The present study revealed a significantly greater incidence of complications in the older patient group (72.7% vs. 39.5%; *p* < 0.05). Operator experience was not correlated with the occurrence of postoperative complications (*p* = 0.531). Postoperative complications were less common after OMIP procedures than after unilateral and bilateral neck exploration (27.3% vs. 78.9%; *p* < 0.05). Additionally, a higher incidence of postoperative complications was observed in patients with comorbidities (*p* < 0.05). Postoperative bleeding was more common in patients with comorbidities (*p* < 0.05). However, comorbidities did not impact the other analyzed complications. The analysis found no statistically significant relationships between the operator’s experience or the patient’s sex and age and the effectiveness of the procedure. All surgical treatment techniques for PHPT demonstrated similar effectiveness. Table 5 summarizes the perioperative data and treatment characteristics.

The most frequently affected glands were the left inferior parathyroid gland (*n* = 65; 43.6%) and the right inferior parathyroid gland (*n* = 43; 29%). The left superior parathyroid gland (*n* = 10; 6.7%) and the right superior parathyroid gland (*n* = 9; 6%) were less frequently affected. An atypical location was identified in 22 (14.7%) of the removed parathyroid glands. The most common atypical location was intrathymic (*n* = 10; 6.6%), followed by the superior mediastinum (*n* = 5; 3.3%), the submandibular region (*n* = 4; 2.7%), and the retroesophageal space (*n* = 2; 1.4%). In one patient (0.7%), the affected parathyroid gland was located within the thyroid gland. No significant correlation was found between the location of the parathyroid adenoma and the patient’s sex. Surgical findings and the location of pathological parathyroid glands are presented in Table 6.

In 101 patients (80.8%), a single parathyroid adenoma was identified, whereas in the remaining 24 patients (19.2%), the pathology involved two glands simultaneously. An adenoma located in the right inferior parathyroid gland was associated with a greater frequency of multiple adenomas (87.5% vs. 12.5%; *p* < 0.05). In most cases, an adenoma in the right inferior parathyroid gland coexisted with an adenoma in the left inferior parathyroid gland (*n* = 8; 33.3%) and less frequently with adenomas of the right superior (*n* = 8; 33.3%), left superior (*n* = 3; 12.5%), and atypical locations (*n* = 2; 8.3%). In the remaining patients, an adenoma of the left inferior parathyroid gland coexisted with an adenoma in an atypical location (*n* = 3; 12.5%). Among older patients, the coexistence of two parathyroid adenomas was significantly more common (62.5% vs. 37.5%; *p* < 0.05). Table 7 provides relations of parathyroid adenomas in discussed multiglandular disease.

The majority of procedures were performed using the OMIP method (*n* = 93; 74.4%), with bilateral neck exploration (*n* = 21; 16.8%) and unilateral neck exploration (*n* = 11; 8.8%) being less common. The average duration of the operations was 53 min, ranging from 35 to 153 min. The analysis revealed a significantly shorter procedure time when conducted by an experienced surgeon than for an operator with less experience (M = 45 min, SD = 12.35 vs. M = 57 min, SD = 25; *p* < 0.05). However, a comparison of the average procedure time for unilateral and bilateral neck exploration did not yield a statistically significant difference (M = 58 min, SD = 26.2 vs. M = 71 min, SD = 27.75; *p* = 0.232). The mean operation time for the OMIP procedure (M = 45 min; SD = 11.9) was significantly shorter than that for the unilateral and bilateral neck exploration techniques (M = 65.8 min; SD = 27.4; *p* < 0.05).

### 3.3. Analysis of Total Serum Calcium and PTH Levels in PHPT

The average total serum calcium levels on the day of admission was 11.59 mg/dL (SD = 1.21), which decreased to 9.37 mg/dL (SD = 1.09) on the first day after surgery (*p* < 0.05). In patients with right inferior parathyroid adenoma, a significantly lower total calcium level was observed on the day of admission than on the first postoperative day following surgical removal (M = 11.31 mg/dL, SD = 1.04 vs. M = 11.77 mg/dL, SD = 1.27; *p* < 0.05) and on the first postoperative day following surgical removal (M = 9.63 mg/dL, SD = 1.13 vs. M = 9.22 mg/dL, SD = 1.04; *p* < 0.05). Conversely, in patients with left inferior parathyroid adenoma, significantly greater total calcium levels were detected on the day of admission (M = 11.91 mg/dL; SD = 1.27 vs. M = 11.79 mg/dL; SD = 0.99; *p* < 0.05). Additionally, the mean total calcium concentration after surgical removal of the left inferior parathyroid gland measured on the first postoperative day was significantly lower than that after surgery involving the other parathyroid glands (M = 9.14 mg/dL; SD = 0.99 vs. M = 9.67 mg/dL; SD = 1.15; *p* < 0.05). The analysis of the mean total calcium level on the day of admission did not reveal a statistically significant difference between patients with adenoma of a single parathyroid gland and those with adenoma of two parathyroid glands (M = 11.57 mg/dL, SD = 1.22 vs. M = 11.82 mg/dL, SD = 0.5; *p* = 0.070). However, the total calcium levels on the first day after surgical removal of adenomas from two parathyroid glands were significantly lower than those on the first day after removal of an adenoma from a single parathyroid gland (M = 8.84 mg/dL, SD = 0.86 vs. M = 9.41 mg/dL, SD = 1.05; *p* < 0.05). The PTH levels in the blood of patients on the day of admission ranged from 61.40 to 1461 pg/mL (M = 330.34 pg/mL; SD = 366.32) and was significantly greater than the PTH levels on the first day after surgery, with values ranging from 5 to 124.60 pg/mL (M = 23.16 pg/mL; SD = 45.09) (*p* < 0.05). There were no statistically significant differences in PTH levels in the blood between different adenoma locations on the day of admission or on the first day after surgery. On the day of admission, patients with a single adenoma had a mean PTH level of 311 pg/mL (SD = 346), whereas patients with two parathyroid adenomas had a mean PTH level of 499.30 pg/mL (SD = 523) (*p* < 0.05). In the majority of patients (*n* = 94; 75.2%), PTH levels in the blood normalized on the first day after the procedure. Normalization of PTH levels was observed within 72 h after the procedure in 12 patients (9.6%) and between the 3rd and 6th day after surgery in 12 patients (9.6%), while 7 patients (5.6%) were discharged from the department with PTH levels in the blood exceeding the upper limit of normal. Patients who underwent removal of an atypical parathyroid adenoma were significantly less likely to achieve normalization of PTH levels on the first day after surgery than were those with typical adenomas (9.3% vs. 35.5%; *p* < 0.05). There was no relationship between PTH normalization and adenomas of the other parathyroid glands. Additionally, surgical removal of co-occurring adenomas from two parathyroid glands was associated with significantly less frequent normalization of PTH levels on the first day after surgery (11.7% vs. 41.9%; *p* < 0.05). The changes in biochemical levels of total serum calcium and PTH in different time intervals are presented in Figure 1.

Upon admission, hypophosphatemia was noted in 66 (53%) patients, with mild occurrences in 55 (82.5%) patients, moderate occurrences in 10 (15%) patients, and severe occurrences in one (1.5%) patient. The grade of hypophosphatemia was categorized as mild with a level of inorganic phosphorus at 3–2.60 mg/dL, as moderate at 2.50–1 mg/dL, and as severe at <1 mg/dL. The mean level of inorganic phosphorus on admission was 2.57 mg/dL (SD = 0.57), whereas on the first postoperative day, it was 3.20 mg/dL (SD = 0.5) (*p* < 0.05). On the first postoperative day, hypocalcemia was observed in 47 (37.6%) patients, with calcium levels in the blood ranging from 5.20 to 8.40 mg/dL. Statistical analysis revealed a significantly greater incidence of hypocalcemia in patients operated on by surgeons who had performed up to 10 parathyroidectomies (50% vs. 30.3%; *p* < 0.05). Decreased total calcium levels were significantly more common in patients who underwent unilateral neck exploration (72.7% vs. 34.2%; *p* < 0.05) and bilateral neck exploration (90.4% vs. 26.9%; *p* < 0.05). However, no statistically significant relationships were found between the incidence of hypocalcemia and patient age or sex. In the study cohort, the average total calcium levels on the day of admission were 11.59 mg/dL (range: 10.74–13.40 mg/dL; SD = 1.21), 9.37 mg/dL (range: 5.20–12.30 mg/dL; SD = 1.09) on the first postoperative day, and 9.13 mg/dL (range: 7.90–14.31 mg/dL; SD = 0.6) six months after the procedure. The calcium levels were significantly lower six months after surgery than they were at admission (*p* < 0.05). However, this study did not reveal a statistically significant difference between the total calcium levels on the first postoperative day and that six months after the procedure (*p* = 0.438). Normalization of total calcium levels six months after the procedure was achieved in 119 (95.2%) patients. Among the six (4.8%) patients whose concentrations were not normalized, the total calcium values ranged from 10.63 to 14.31 mg/dL (M = 12.40 mg/dL, SD = 0.95). Finally, the serum calcium and PTH levels were also related to the volume of the parathyroid adenoma. The mean volume of the removed parathyroid gland was 1.9 cm^3^ (range 0.4–4.05 cm^3^). The study showed a statistically significant correlation between adenoma volume and serum PTH levels (*p* < 0.05), but there was no relationship with the extent of changes in PTH levels on the first postoperative day (*p* = 0.110) (Figure 2).

However, no significant effect of parathyroid adenoma size on total serum calcium levels was observed (*p* = 0.873), also with no dependence between these variables in the first day after parathyroidectomy (*p* = 0.071) (Figure 3).

## 4. Discussion

Despite a noticeable increase in awareness and greater interest in the field of PHPT, with a significantly growing number of publications, many authors emphasize the underdiagnosis of this disease [17,18,19]. Owing to the lack of characteristic symptoms and the peak incidence of the disease in the postmenopausal period, PHPT poses a challenge to its diagnosis. Clinicians may disregard elevated serum calcium levels and ignore PHPT as a potential cause of symptoms, leading to delayed diagnosis and treatment [20,21]. Epidemiological studies of the European population indicate that the increased availability of tests for total calcium and PTH levels in serum has a positive effect on increased detection of the disease [22,23]. An analysis of epidemiological studies based on the American population highlights the importance of assessing calcium and PTH levels in asymptomatic patients [20,24].

Our study included patients who underwent surgery for primary PHPT between 2008 and 2017 at one center, comprising 110 (88%) women and 15 (12%) men. Research has confirmed that female sex predisposes individuals to the development of PHPT, especially at postmenopausal ages [25]. The reduced antiproliferative effect of estrogens on parathyroid cells plays an important role in this regard [26]. Women receiving hormone replacement therapy have a reduced risk of developing PHPT [27]. This investigation revealed intriguing insights into the relationships between blood calcium levels and various symptoms associated with PHPT. Notably, a statistically significant association was found between elevated blood calcium levels and specific symptoms reported by patients. For example, individuals who experienced osteoarticular pain presented higher blood calcium levels than did those without this symptom (12.05 mg/dL vs. 11.03 mg/dL; *p* < 0.05). Similarly, patients reporting muscle weakness or osteoporotic changes also presented elevated blood calcium levels (11.73 mg/dL vs. 11.03 mg/dL; *p* < 0.05 and 12.05 mg/dL vs. 11.02 mg/dL; *p* < 0.05, respectively). Furthermore, this study revealed a notable correlation between PTH levels and osteoporosis, highlighting an interplay between hormonal regulation and skeletal health (356.03 pg/mL vs. 297.64 pg/mL; *p* < 0.05). However, Chan et al.’s analysis revealed contrasting findings showing no relationship between the levels of total calcium and PTH in the blood and symptoms of the musculoskeletal system [28]. Interestingly, sex did not emerge as a significant factor in reported PHPT symptoms, underscoring the multifactorial nature of the condition. Nonetheless, age proved to be a pertinent determinant, with older individuals exhibiting a greater prevalence of muscle weakness (98.1% vs. 51.3%; *p* < 0.05), osteoarticular pain (86.7% vs. 31.9%; *p* < 0.05), and osteoporosis (86.7% vs. 33.3%; *p* < 0.05) than their younger counterparts.

The psychiatric sequelae observed in patients with PHPT may stem from various physiological mechanisms. These include heightened neurotransmitter transmission due to diminished monoamine oxidase (MAO) activity, impaired functioning of the ATP-dependent sodium-potassium pump (Na^+^/K^+^ ATPase), and elevated Ca^2+^ concentration within synapses [29]. While the literature generally indicates a lack of an association between psychological symptomatology in patients with PHPT and serum calcium levels, certain studies suggest a heightened risk of severe psychotic disorders in patients whose hypercalcemia exceeds 14 g/dL [3,30,31]. In our study, depressive mood was prevalent among 103 (82.4%) patients. However, our analysis revealed no significant correlation between the occurrence of depressive mood and serum calcium or PTH levels, the number of pathologically altered parathyroid glands, or the age and sex of the patients. These findings underscore the complexity of psychiatric manifestations in PHPT, suggesting that factors beyond calcium and PTH levels may contribute to the development of depressive symptoms. Further research is warranted to elucidate the underlying mechanisms and optimize therapeutic approaches for psychiatric sequelae in patients with PHPT.

PHPT is commonly associated with an elevated risk of cholelithiasis, affecting approximately 22–30% of patients [32,33]. Saito et al. and Broulik et al. proposed a model in which the key element is the impact of PTH and hypercalcemia on gallbladder contractility impairment, leading to an increased calcium concentration in bile secretions and slowed bile flow in ducts. Furthermore, the authors highlight a significantly higher prevalence of gallstones among women. This sexual dimorphism is attributed to the influence of estrogen, which enhances cholesterol absorption from the digestive system, promotes cholesterol secretion into bile, and inhibits deoxycholic acid synthesis [34,35]. Broulik et al. corroborated a more frequent occurrence of cholelithiasis in patients with PHPT and reported a positive correlation between the incidence of cholelithiasis and advanced age. In our investigation, cholelithiasis was documented in 68 (54%) patients, with no discernible difference in occurrence between women and men (55.4% vs. 46.6%; *p* = 0.521). However, a significantly greater prevalence of cholelithiasis was noted in older patients (86.7% vs. 30.5%; *p* < 0.05). Moreover, patients with cholelithiasis presented significantly elevated levels of both calcium (12.03 mg/dL vs. 11.06 mg/dL; *p* < 0.05) and PTH (357 pg/mL vs. 298.52 pg/mL; *p* < 0.05).

Patients with PHPT face a substantial risk, ranging from 40% to 60%, of developing renal complications, with urinary tract stones being the most prevalent among them [2]. The precise etiopathology underlying the formation of renal and urinary tract deposits remains elusive; however, hypercalciuria has been identified as a key risk factor [36]. Clinical investigations commonly report a heightened incidence of urolithiasis in male patients with PHPT under the age of 50 [36,37]. In contrast to findings from previously published studies, our research did not reveal any correlation between patient sex or age and the risk of urolithiasis development. Nevertheless, individuals afflicted with urolithiasis presented significantly elevated concentrations of total calcium (12.03 mg/dL vs. 11.02 mg/dL; *p* < 0.05) and PTH (356.90 pg/mL vs. 296.42 pg/mL; *p* < 0.05), mirroring the observations reported by Corbett et al. [38]. These findings underscore the complexity of the relationship between PHPT and renal complications, emphasizing the multifaceted nature of disease progression.

The primary cause of extended hospital stays following parathyroidectomy is often severe hypocalcemia [39]. Lansdown et al. conducted an analysis of 17,498 patients who underwent surgery between 2014 and 2019 and reported a shorter duration of hospitalization for patients operated on by surgeons who had performed more than 60 parathyroidectomies than for those who had performed fewer than 10 procedures [40]. Additionally, their study highlighted the influence of surgeon experience on various outcomes, including a lower percentage of readmissions within 30 days post-procedure (11.7% vs. 7%), a decreased incidence of postoperative hypoparathyroidism (16.3% vs. 10.9%), fewer instances of PHPT recurrence within a year of surgery (2.5% vs. 1%), and reduced postoperative mortality (1% vs. 0.5%). Conversely, research by Thomas et al., which included 7313 patients, identified a heightened risk of prolonged hospitalization among individuals aged over 65 years [41]. In our study, the hospitalization period ranged from 3 to 16 days, with a median of 4.40 days. The longest hospitalization, which lasted 16 days, involved an 82-year-old patient who experienced cardiopulmonary failure in the postoperative period. Among the 11 patients who experienced postoperative complications, the duration of hospitalization was significantly longer than that in the remainder of the cohort (9.30 days vs. 3.92 days; *p* < 0.05). Similarly, patients aged 65 and older experienced prolonged hospital stays (5.16 days vs. 3.90 days; *p* < 0.05). However, our investigation did not identify any disparities in hospitalization duration on the basis of surgeon experience.

Our investigation revealed that the most frequent site of adenoma occurrence was the left inferior parathyroid gland (*n* = 65; 43%), with an atypical location observed in 22 (14.7%) of the excised parathyroid glands, predominantly involving intrathymic sites (*n* = 10; 6.6%). No statistically significant differences were found between adenoma location and sex. Analysis of laboratory parameters revealed a notably lower calcium level on admission in patients with right inferior parathyroid adenomas than in those with other parathyroid glands, along with significantly higher calcium levels on the first postoperative day (*p* < 0.05). Similarly, significantly elevated calcium levels on admission were observed in patients with left inferior parathyroid adenomas compared with those with adenomas affecting other glands, with a corresponding decrease on the first postoperative day (*p* < 0.05). No statistically significant differences were noted in calcium levels in patients with adenomas affecting other parathyroid glands, and no correlation was found between PTH levels and adenoma location. However, a significant increase in the mean inorganic phosphate level was observed on the first postoperative day following parathyroid adenoma removal (2.57 mg/dL vs. 3.20 mg/dL; *p* < 0.05). A negative prognostic indicator is the concurrent presence of adenomas affecting two or more parathyroid glands. According to the available literature, the prevalence of dual adenomas ranges from 1% to 10%, with women and individuals over 60 years of age being at increased risk [42]. Surprisingly, in our study, a substantial number of patients had dual parathyroid adenomas (*n* = 24; 19.2%). The most frequent scenario involved the coexistence of a right inferior parathyroid adenoma with a left inferior one (*n* = 8; 33%) and a right inferior one with a left superior one (*n* = 8; 33%). There was no statistically significant disparity in the occurrence of dual parathyroid adenomas between sexes; however, their incidence was notably greater in the older patient group (62.5% vs. 37.5%; *p* < 0.05). With respect to calcium levels at admission, no statistically significant differences were detected between patients with adenomas affecting two parathyroid glands and those with adenomas of a single gland. Nonetheless, calcium levels were significantly lower on the first post-procedure day in patients with pathology involving two glands (8.84 mg/dL vs. 9.41 mg/dL; *p* < 0.05). Furthermore, patients with dual adenomas presented significantly greater PTH levels at admission and on the first postoperative day.

The current gold standard for treating PHPT is surgery [8,9]. Depending on the center’s expertise and technical capabilities, surgical interventions are conducted in a minimally invasive manner or via traditional unilateral or bilateral neck exploration. Surgical management of PHPT, irrespective of the chosen approach, has a high efficacy rate ranging from 97% to 99% [43]. In our investigation, 93 (74.4%) patients underwent OMIP procedures, 11 (8.8%) underwent unilateral neck exploration, and 21 (16.8%) underwent bilateral neck exploration. The mean operation duration was 53 min. Our findings revealed a significantly shorter duration for OMIP procedures than for both unilateral and bilateral neck explorations (45 min vs. 65.80 min; *p* < 0.05), with no notable disparity between the two neck exploration techniques. Consistent with findings from other studies [44,45,46], our research demonstrated a significantly shorter procedure duration with the minimally invasive approach and no significant time discrepancies between unilateral and bilateral neck explorations. Additionally, in our study, procedure duration was significantly correlated with operator experience (45 min vs. 57 min; *p* < 0.05).

The incidence of complications varies significantly in the available literature (1–59.7%) and largely depends on whether the study investigators included transient hypocalcemia as a postoperative complication [47,48,49]. In our study, postoperative hypocalcemia was observed in 47 (37.6%) patients, with calcium levels ranging from 5.20 to 8.90 mg/dL. The patient with extremely low serum calcium values (5.20 mg/dL) presented symptoms of tetany. Additionally, two (1.6%) patients reported symptoms of paresthesia in the form of numbness in the fingers of the upper limbs and tingling around the mouth. In each case, intravenous infusions of calcium chloride (CaCl_2_) and magnesium sulfate (MgSO_4_) were administered, resulting in the disappearance of symptoms. Our observations were consistent with the findings of other authors, demonstrating no correlation between patient age and the occurrence of postoperative complications [50]. However, hypocalcemia occurred more frequently in the group of patients who underwent surgery by a surgeon with less experience (50% vs. 30.3%; *p* < 0.05). Differences between treatment techniques were also significant. Hypocalcemia occurred least frequently after OMIP procedures (21.5% vs. 84.3%; *p* < 0.05) and most frequently after bilateral neck exploration (90.4% vs. 26.9%; *p* < 0.05). Among the remaining complications, four (3.2%) cases of postoperative wound infection and three (2.4%) cases of postoperative bleeding were observed. Symptoms of cardiorespiratory failure were noted in one (0.8%) patient. This patient required treatment in the intensive care unit. In our study, no relationship between the patient’s sex or age and the risk of complications was observed. There was no relationship between the operator’s experience or the location of the pathologically changed parathyroid gland and the occurrence of complications during the postoperative period. However, the influence of comorbidities on the increased risk of complications (20.9% vs. 2.4%; *p* < 0.05) included a greater risk of bleeding in the postoperative period (6.9% vs. 0%; *p* < 0.05). The mean volume of the resected parathyroid gland was 1.9 cm^3^ (range, 0.4–4.05 cm^3^). Our findings indicate that larger adenoma size may be associated with higher preoperative serum PTH levels. Similar observations were reported by Ramas et al. and Rezkallah et al., who additionally noted that this correlation could enable a more accurate estimation of gland volume, potentially minimizing unnecessary tissue dissection [51,52]. In contrast to other studies, our analysis did not demonstrate a significant relationship between parathyroid adenoma volume and total serum calcium levels. Furthermore, we did not detect a statistically significant influence of resected adenoma volume on the size of PTH and total calcium levels reduction on the first day after surgery. Nevertheless, several published reports have highlighted that removal of larger parathyroid lesions may precipitate a marked decrease in PTH and total calcium levels, leading to clinical manifestations of hypocalcemia [53,54].

Successful parathyroidectomy is a procedure after which hypercalcemia does not develop within a period of six months [55,56]. In this study, normalization of calcium levels in the sixth month after surgery was achieved in 119 (95.2%) patients. Among the six patients whose calcium levels remained above 10.50 g/dL, five underwent surgery via the OMIP technique, whereas one patient underwent unilateral neck exploration. The normalization of PTH levels after parathyroidectomy represents an additional concern. In our study, the vast majority of patients had normalized PTH levels up to a maximum of day 6 after the procedure, but in seven patients (5.6%), elevated PTH levels were maintained after hospitalization. This condition may result from advanced patient age, impaired renal function, or exceptionally large parathyroid adenomas and may affect up to one-third of patients undergoing surgical treatment for parathyroid adenomas. However, this is generally not associated with hypercalcemia and is considered a successful parathyroidectomy, but these cases require long-term postoperative monitoring and extended diagnostic imaging to detect potential adenoma recurrence [57,58]. The association of calcium and PTH levels with alkaline phosphatase is also an important issue. Alkaline phosphatase is a bone formation marker secreted by osteoblasts, with usually elevated levels in PHPT. High levels of alkaline phosphatase are identified as a predictor of calcium and PTH decline that correlates with postoperative hypocalcemia. After parathyroidectomy, there is a significant disconnect between the processes of bone formation and resorption, which is affected by a rapid decrease in PTH and high levels of alkaline phosphatase, which increases the risk of hypocalcemia. The decrease in calcium levels may also be lower in patients with lower alkaline phosphatase levels, but such a correlation can be found with high preoperative calcium levels. Patients with different preoperative serum calcium levels vary in factors affecting hypocalcemia, with alkaline phosphatase possibly representing one of these factors [59,60,61]. However, such analysis is beyond the scope of this study. No relationship between the effectiveness of the procedure and patient sex or age was found. There were no significant differences in effectiveness between individual parathyroidectomy techniques or in the location of the surgically removed adenoma.

## 5. Conclusions

PHPT was effectively treated with parathyroidectomy, irrespective of the patient’s age, sex, or location of the affected gland. Surgical removal of a parathyroid adenoma led to a reduction in serum PTH and total calcium levels, along with an increase in inorganic phosphate level. The left inferior parathyroid gland was the most common site of adenoma occurrence; thus, this site is advised to be primarily assessed during bilateral neck exploration. Elevated serum PTH levels may have indicated the presence of multiple adenomas, warranting consideration for bilateral neck exploration. Patients with two concurrent parathyroid adenomas required close monitoring postoperatively due to a more pronounced decrease in calcium levels. Additionally, patients with comorbidities should undergo intensified observation postoperatively due to the increased risk of complications.

## Figures and Tables

**Figure 1 jcm-14-00996-f001:**
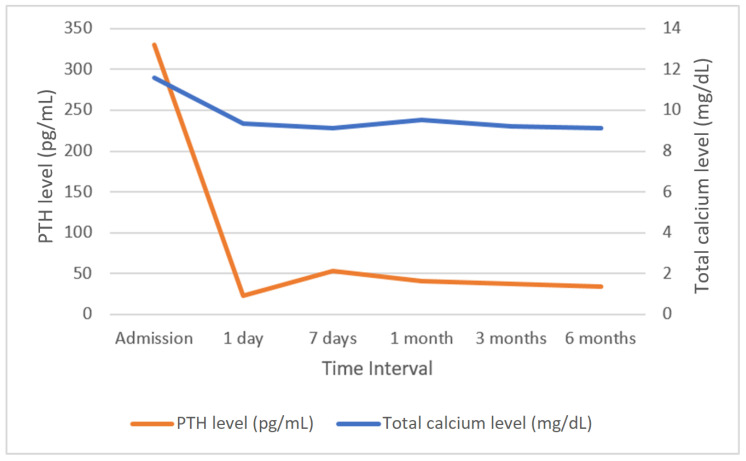
Fluctuations in total calcium (blue line) and PTH (orange line) levels measured on the day of admission and at subsequent follow-up points. PTH—parathormone.

**Figure 2 jcm-14-00996-f002:**
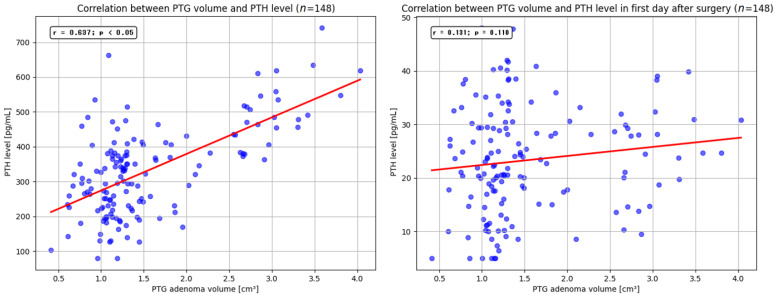
Correlation between PTG volume and PTH levels in admission day and after parathyroidectomy. PTG—parathyroid gland; PTH—parathormone.

**Figure 3 jcm-14-00996-f003:**
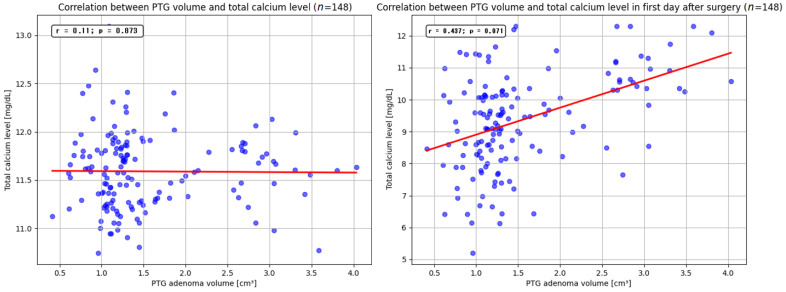
Correlation between PTG volume and total calcium levels in admission day and after parathyroidectomy. PTG—parathyroid gland; PTH—parathormone.

**Table 1 jcm-14-00996-t001:** Patient characteristics.

Feature	*n* (%)
**Sex**	
Female	110 (88)
Male	15 (12)
Total	125 (100)
**Age**	
≥65 years old	53 (42.4)
<65 years old	72 (57.6)
**Comorbidity**	
Hypertension	27 (21.6)
Diabetes	15 (12)
Gout	7 (5.6)
Chronic obstructive pulmonary disease	4 (3.2)
Kidney failure	4 (3.2)
Other cardiovascular diseases	10 (12.5)
**Symptoms of PHPT**	
Depressed mood	103 (82.4)
Nausea	102 (81.6)
Headache	102 (81.6)
Weakened muscle strength	89 (71.2)
Urinary tract stones	70 (56)
Osteoporosis	70 (56)
Osteoarticular pain	69 (55.2)
Cholelithiasis	68 (54)
Other	12 (9.6)

**Table 2 jcm-14-00996-t002:** Relationship between the occurrence of PHPT symptoms and age. A *p* < 0.05 indicates a statistically significant value.

Feature	Patient Age < 65 (*n* = 72)	Patient Age ≥ 65 (*n* = 53)	*p* Value
**Depressed mood**			
Yes	61 (84.7)	42 (79.2)	0.426
No	11 (15.3)	11 (20.8)	
**Nausea**			
Yes	62 (83.3)	42 (79.2)	0.559
No	12 (16.7)	11 (20.8)	
**Headache**			
Yes	59 (81.9)	43 (81.1)	0.907
No	13 (18.1)	10 (18.9)	
**Muscle weakness**			
Yes	37 (51.3)	52 (98.1)	<0.05
No	35 (48.7)	1 (1.9)	
**Urinary tract stones**			
Yes	24 (33.3)	46 (86.7)	<0.05
No	48 (66.6)	7 (13.2)	
**Osteoporosis**			
Yes	24 (33.3)	46 (86.7)	<0.05
No	48 (66.6)	7 (13.2)	
**Osteoarticular pain**			
Yes	23 (31.9)	46 (86.7)	<0.05
No	49 (68.1)	7 (13.3)	
**Cholelithiasis**			
Yes	22 (30.5)	46 (86.7)	<0.05
No	50 (69.5)	7 (13.3)	

**Table 3 jcm-14-00996-t003:** The relation between the occurrence of PHPT symptoms and total serum calcium level. A *p* < 0.05 indicates a statistically significant value.

Symptoms	*n* (%)	Mean Total Calcium Level [mg/dL] (SD)	*p* Value
**Depressed mood**			
Yes	103 (82.4)	11.57 (1.57)	0.666
No	22 (17.6)	11.60 (0.88)	
**Nausea**			
Yes	102 (81.6)	11.56 (1.26)	0.454
No	23 (18.4)	11.72 (0.89)	
**Headache**			
Yes	102 (81.6)	11.60 (1.28)	0.800
No	23 (18.4)	11.60 (0.76)	
**Muscle weakness**			
Yes	89 (71.2)	11.73 (1.15)	<0.05
No	36 (28.8)	11.03 (1.25)	
**Urinary tract stones**			
Yes	70 (56)	12.03 (1.06)	<0.05
No	55 (44)	11.02 (1.13)	
**Osteoporosis**			
Yes	70 (56)	12.05 (1.06)	<0.05
No	55 (44)	11.01 (1.12)	
**Osteoarticular pain**			
Yes	69 (55.2)	12.05 (1.12)	<0.05
No	56 (44.8)	11.03 (1.06)	
**Cholelithiasis**			
Yes	68 (54)	12.03 (1.06)	<0.05
No	57 (46)	11.06 (1.14)	

**Table 4 jcm-14-00996-t004:** The relation between the occurrence of PHPT symptoms and PTH level in the blood. A *p* < 0.05 indicates a statistically significant value.

Symptoms	*n* (%)	Mean Serum PTH Level [pg/mL] (SD)	*p* Value
**Depressed mood**			
Yes	103 (82.4)	341.24 (379.18)	0.320
No	22 (17.6)	279.28 (301.12)	
**Nausea**			
Yes	102 (81.6)	360.12 (348.2)	0.454
No	23 (18.4)	336.20 (315)	
**Headache**			
Yes	102 (81.6)	342.04 (380.3)	0.298
No	23 (18.4)	278.43 (298.12)	
**Muscle weakness**			
Yes	89 (71.2)	316.78 (319.46)	0.749
No	36 (28.8)	363.85 (466.03)	
**Urinary tract stones**			
Yes	70 (56)	356.90 (348)	<0.05
No	55 (44)	296.42 (388.9)	
**Osteoporosis**			
Yes	70 (56)	356.03 (348.6)	<0.05
No	55 (44)	297.64 (388.4)	
**Osteoarticular pain**			
Yes	69 (55.2)	353.05 (350.62)	0.070
No	56 (44.8)	302.35 (386.12)	
**Cholelithiasis**			
Yes	68 (54)	357 (351)	<0.05
No	57 (46)	298.52 (383.7)	

**Table 5 jcm-14-00996-t005:** Characteristics of the perioperative data, surgical complications and patient hospitalization time.

Surgery Type	*n* (%)
OMIP	93 (74.4)
Bilateral neck exploration	21 (16.8)
Unilateral neck exploration	11 (8.8)
Total	125 (100)
**Surgical complicatons**	***n* (%)**
Wound infection	4 (36)
Postoperative bleeding	3 (27)
Parasthesia	2 (18)
Tetany	1 (9)
Cardiopulmonary failure	1 (9)
Total	11 (100)
**Operative time**	**min (M)**
OMIP	35–100 (41)
Bilateral neck exploration	45–153 (62.50)
Unilateral neck exploration	42–131 (46)
Overall	35–153 (53)
**Hospitalization time**	**days (M)**
≥65 years old	3–16 (5.19)
<65 years old	3–10 (3.90)
Overall	3–16 (4.40)

OMIP—open minimally invasive parathyroidectomy.

**Table 6 jcm-14-00996-t006:** Summary of pathological parathyroid glands and their location.

Number of Pathological PTG	*n* (%)
Single	101 (80.8)
Multiple	24 (19.2)
Total	125 (100)
**Location of pathological PTG**	***n* (%)**
Right superior	9 (6)
Right inferior	43 (29)
Left superior	10 (6.7)
Left inferior	65 (43.6)
Atypical	22 (14.7)
Total	149 (100)
**Atypical location**	***n* (%)**
Intrathymic	10 (45.5)
Upper mediastinal	5 (22.7)
Submandibular	4 (18.2)
Retroesophageal	2 (9.1)
Intrathyroidal	1 (4.5)
Total	22 (100)

PTG—parathyroid gland.

**Table 7 jcm-14-00996-t007:** Details about multiple pathological parathyroid location.

Location	Right Superior	Right Inferior	Left Superior	Left Inferior	Atypical	Total, *n* (100)
**Right superior**	–	3	0	0	0	3 (6.3)
**Right inferior**	3	–	8	8	2	21 (43.8)
**Left superior**	0	8	–	0	0	8 (16.7)
**Left inferior**	0	8	0	–	3	11 (22.9)
**Atypical**	0	2	0	3	–	5 (10.3)
**Total**	3	21	8	11	5	48 (100)

## Data Availability

The datasets used and/or analyzed during the current study are available from the authors upon reasonable request.

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
