# Peer review of "Analysis of Clinical and Biochemical Parameters and the Effectiveness of Surgical Treatment in Patients with Primary Hyperparathyroidism: A Single-Center Study"

_jcm, 2025, doi:10.3390/jcm14030996_

Round 1
Reviewer 1 Report
Comments and Suggestions for Authors
Major Comments
- Definition of Symptoms
Please specify the method used to assess and define the symptoms related to hyperparathyroidism. Were they evaluated using a questionnaire or through objective measurements? At what time points were these symptoms assessed? For example, how did you define depressed mood or weakened muscle strength? Additionally, what were the specific symptoms of the 12 patients categorized as "other"? Since these symptoms are a major outcome in your study, the lack of clear definitions severely compromises the reliability of the results. - Presentation of Results
The presentation of results should be critically revised for clarity and readability. For example: - a. Include one or two tables summarizing the perioperative data, such as the number of parathyroid glands removed per patient (e.g., one gland, two glands, or three glands, respectively), the location of the glands (laterality, superior, or inferior), and complications (e.g., hoarseness, bleeding, seroma, hungry bone syndrome). Also, include data on hospital stay to help readers better understand the study population.
- b. Provide a figure or table to show the biochemical changes in PTH, calcium, and phosphate at different time intervals (e.g., postoperative day 1, one week, one month).
- c. Add a table detailing the locations of glands in patients with multiglandular disease.
- d. The manuscript lacks appropriate paragraphing, making it difficult to read. Consider adding subheadings to improve structure and readability.
- Correlation with Specimen Weight and Alkaline Phosphatase
Literature indicates that the extent of postoperative PTH/calcium drop is associated with specimen weight and serum levels of alkaline phosphatase. Please provide these data for correlation. - Postoperative PTH Levels
The manuscript states that PTH levels normalized in 94 patients on postoperative day 1 and in 12 patients on day 3. Additionally, 7 patients still had elevated PTH levels after discharge. What were the PTH levels of the remaining 12 patients (125 - 94 - 12 - 7 = 12)? What are the potential causes for patients whose PTH levels remained above the normal range?
Minor Comments
- Inconsistent P-Value Reporting
The expression of p-values in Tables 2-4 is inconsistent. For instance, "P < 0.001," "p < 0.005," and "p < 0.05" are all used to indicate statistical significance. Please standardize the reporting format. - Definition of Hypophosphatemia Severity
In lines 244-246, what are the definitions of mild, moderate, and severe hypophosphatemia? Please clarify.
Reviewer 2 Report
Comments and Suggestions for Authors
Dear authors,
I would thank you for preparing this interesting article, below I am presenting some remarks and comments:
Line 16/17 please consider changing sentence to less general, e.g. mentioning Ca-P
Line 19/20 the sentence is poorly formulated, there is no cause-outcome correlation
Consider using “left/right inferior PA” not upper/lower
Line 75 Radioisotopes should be presented as the guidelines states, which is: [99mTc]Tc-MIBI
Same for [11C]C-methionine and [18F]F-choline
Table 3 & 4 - please consider changing correlation to relation, as in statistics correlation is more associated with Pearson or Spearman tests
Why statistically significant values are written as <.005 and <0.05 ; while non-significant have exact values? Please consider unification (I recommend all p-values written as exact results e.g. 0.543; 0.123, 0.032, 0.001, or <0.001.
Similar concern about laboratory results (e.g. LINE 165 – why results do not have the same decimal scale? - 356.9 pg/ml vs. 296.42 pg/ml) please be consistent and unify (eg. 356.90 vs 296.42 or 356.9 vs 296.4)
DISCUSSION
Line 383- 395. Please consider not mixing results of your study with discussion.
Conclusions:
PHPT presents a diagnostic challenge for many clinicians. – this is not a conclusion directly resulting from the study
Please consider using past tense in describing conclusions, e.g.
“left lower parathyroid gland is the most common site of adenoma occurrence..” à
“left lower parathyroid gland was the most common site of adenoma occurrence, thus this site is advised to be primarily assessed during bilateral neck exploration.”
Round 2
Reviewer 1 Report
Comments and Suggestions for Authors
The authors addressed all queries